# Identification of Key Genes in the Synthesis Pathway of Volatile Terpenoids in Fruit of *Zanthoxylum bungeanum* Maxim

**Jingwei Shi [1,2], Xitong Fei [1,2], Yang Hu [1,2], Yulin Liu [1,2] and Anzhi Wei [1,2,*]**

[1]    College of Forestry, Northwest Agriculture and Forestry University, Yangling District, Xianyang 712100, China; shijingweijw@163.com (J.S.); feixt666@163.com (X.F.); huyang19960312@163.com (Y.H.); lyl12504001@126.com (Y.L.)

[2]    Research Centre for Engineering and Technology of Zanthoxylum State Forestry Administration, Yangling District, Xianyang 712100, China

[*]    Correspondence: weianzhi@nwafu.edu.cn; Tel.: +86-029-8708-2211

**Abstract:** *Zanthoxylum bungeanum* Maxim. (*Z. bungeanum*), a plant that belongs to the Rutaceae family, is widely planted in China. Its outstanding feature is its rich aroma. The main component that creates this aroma is the volatile terpenoids. In this study, we aimed to understand the molecular mechanism related to aroma synthesis in *Z. bungeanum* and provide new ideas for breeding. Headspace solid phase micro extraction-gas chromatography mass spectrometry (HS-SPME-GC-MS), RT-qPCR and bioinformatics were used to study the changes in volatile terpenoids and identify key genes in the pathway of terpenoids in fruits of *Z. bungeanum*. The results show that the trend of volatile terpenoids is consistent among the two varieties. As the fruit matures, the terpenoids gradually accumulate and peak in the third period (mid-development) before gradually decreasing. Among these terpenoids, there is the highest content of α-pinene. In *Z. bungeanum* cv. 'Hanchengdahongpao' (Hanchengdahongpao) and *Z. bungeanum* cv. 'Fuguhuajiao' (Fuguhuajiao), this reached 24.74% and 20.78% respectively. In general, for the content of volatile terpenoids, Hanchengdahongpao is 62% and Fuguhuajiao is 41%. The results of RT-qPCR showed that most gene expression in this study was upregulated. Among them, *ZbDXS* has the highest relative level of expression in itself, which is the key rate-limiting enzymein the MEP pathway. These results explore the synthetic route of terpenoids during the ripening process of *Z. bungeanum*, which provides a reference for cultivar and improving good traits.

**Keywords:** *Zanthoxylum bungeanum* Maxim.; aroma; terpenoid; bioinformatics analysis; developmental stages

## 1. Introduction

*Zanthoxylum bungeanum* Maxim. (*Z. bungeanum*) belongs to the Rutaceae family and is an important medicinal and edible plant. China is the largest area of *Z. bungeanum* cultivation in the world [1]. *Z. bungeanum* is also colloquially known as Chinese prickly ash, which is usually divided into two cultivars according to the color of the fruit: *Zanthoxylum Bungeanum* (Red huajiao) and *Zanthoxylum armatum* (Green huajiao) [2]. The main features of *Z. bungeanum* include its numbness and aroma. Furthermore, they are widely used as seasoning in various cuisines, especially Sichuan cuisine, which has a unique flavor [3]. Aroma is one of the most important indexes to evaluate the quality of *Z. bungeanum*, and the main component of aroma is volatile terpenoid [4]. Plants produce a considerable number of volatile organic compounds (VOCs) during their growth, which is a response to changes in

the environment [5]. In addition, VOCs are also widely used in the food, pharmaceutical and skin care industries [6]. They are also developed as a biofuel due to their good prospects and utilization value [7].

The VOCs of plants mainly include terpenoids, phenylpropanes and aliphatic derivatives. Terpenoids are usually found in higher plants that are widely used in cosmetics and medicine [8]. It is the largest group involved in the secondary metabolism of plants and its species are sesquiterpenes, monoterpenes, diterpenes, triterpenes, and polyterpenes [9]. Terpenoids are the main components of most fruit aromas. In *citrus*, terpenoids are the main component of their aroma [10]. Furthermore, for *Vitis vinifera* L., terpenes are the main aroma components and Okamoto isolated 36 monoterpenes from *Vitis vinifera* L. [11]. It is also the main component of floral fragrance, which is closely related to plant pollination [7]. Experiments with the bumblebee showed that the monoterpenoids released by the monkeyflower played an important role in pollination [12]. In general, terpenoids are widely used and need to be explored further.

There are two main synthetic pathways of terpenes in plants. The first is the mevalonic acid (MVA) pathway, which occurs in the cytosol. The other one occurs in plastids and is called the methylerythritol phosphate (MEP) pathway [13] (Figure 1). *DXS* and *HMGR* are the key rate-limiting enzymes of the MEP pathway and the MVA pathway, respectively [14,15]. These two pathways occur independently in different spaces, but they both produce isopentenyl diphosphate (IPP) precursors of terpenoids [16]. After the synthesis of precursors of terpenes, the downstream terpenes synthase plays an important role in catalyzing geranyl pyrophosphate (GPP), farnesyl pyrophosphate (FPP) and geranylgeranyl pyrophosphate (GGPP) to synthesize monoterpenes, sesquiterpene and diterpene, respectively [9,17,18].

In the MVA pathway, Acetyl-CoA (AC) is first obtained by immobilizing $CO_2$ in the cytoplasm before the two molecules of AC are condensed into one molecule of acetoacetyl-CoA (ACC) by acetyl-CoA C-acetyltransferase (ACA) catalysis [19]. Hydroxymethylglutaryl-CoA synthase (HCS) is catalyzed into hydroxymethylglutaryl-CoA reductase (HMGR) by AC and ACA [20]. After this, HMGR is converted to mevalonate (MVA) [21]. MVA further produces phosphomevalonate kinase (MVAP), which undergoes another enzymatic reaction to form isopentenyl pyrophosphate (IPP) [22]. IPP forms farnesyl pyrophosphate (FPP) by farnesyl diphosphate synthase (ZFPS), which further generates sesquiterpene. The MEP approach is roughly described as the enzymatic reaction of 1-deoxy-D-xylulose-5-phosphate synthase (DXS), which is condensed into 1-deoxy-D-xylulose-5-phosphate (DXP) [23]. DXP is reduced to 2-C-Methyl-D-erythritol-4-phosphate (MEP) by 1-deoxy-D-xylulose-5-phosphate reductoisomeras (DXR) [24]. MEP is condensed into 2-Phospho-4-(cytidine 5′-diphospho)-2-C-methyl-D-erythrito-l by 4-diphosphocytidyl-2-C-methyl-D-erythritol kinase (ISPE). There is the subsequent production of 2-C-Methyl-D-erythritol 2,4-cyclodiphosphate (CMDCP) by 2-C-methyl-D-erythritol 2,4-cyclodiphosphate synthase (ISPF). Being catalyzed by 2-C-methyl-D-erythritol 2,4-cyclodiphosphate synthase (GCPE), CMDCP forms 1-Hydroxy-2-methyl-2-butenyl-4-diphosphate. Finally, this is converted to IPP and DMAPP by IPP/DMAPP synthase (IDS) to generate geranylpyrophosphate (GPP), which is used to further generate monoterpenes [25].

At the molecular level, studying the molecular mechanism of volatile terpenoids is an important way to understand the mechanism of fruit aroma formation, which has been extensively studied in many species. Negre and Dudareva's experiments with aroma have revealed that aroma may be related to key gene expression [26,27]. In *Arabidopsis thaliana*, the combined action of gibberellin and jasmonic acid can regulate the synthesis of sesquiterpenes [28]. In *Rosa rugosa* Thunb., Nudix hydrolase RhNUDX1 regulates the biosynthesis of monoterpene alcohols, thereby regulating the aroma of *Rosa rugosa* Thunb. [29]. The terpene synthase (TPS) gene is vital for VOC regulation in plants and can regulate its release under specific space and time conditions [30]. However, in *Z. bungeanum*, the research on aroma mainly focuses on the determination of aroma composition and content. There are many gaps in molecular mechanism research, which need to be urgently addressed.

In the present study, two varieties of *Z. bungeanum*: Hanchengdahongpao and Fuguhuajiao were used as experimental materials. We analyzed volatile organic compounds in *Z. bungeanum* by HS-SPME-GC-MS. We designed primers based on the conserved sequence alignment and investigated

the expression profiles of candidate genes in the metabolic pathway by RT-qPCR. Subsequently, we combined these results with bioinformatics to analyzethe identified key genes. Based on the above analysis results, we provided a reference for the terpenoid biosynthesis of *Z. bungeanum*. At the same time, it provides a scientific basis for controlling aroma and improving fruit quality.

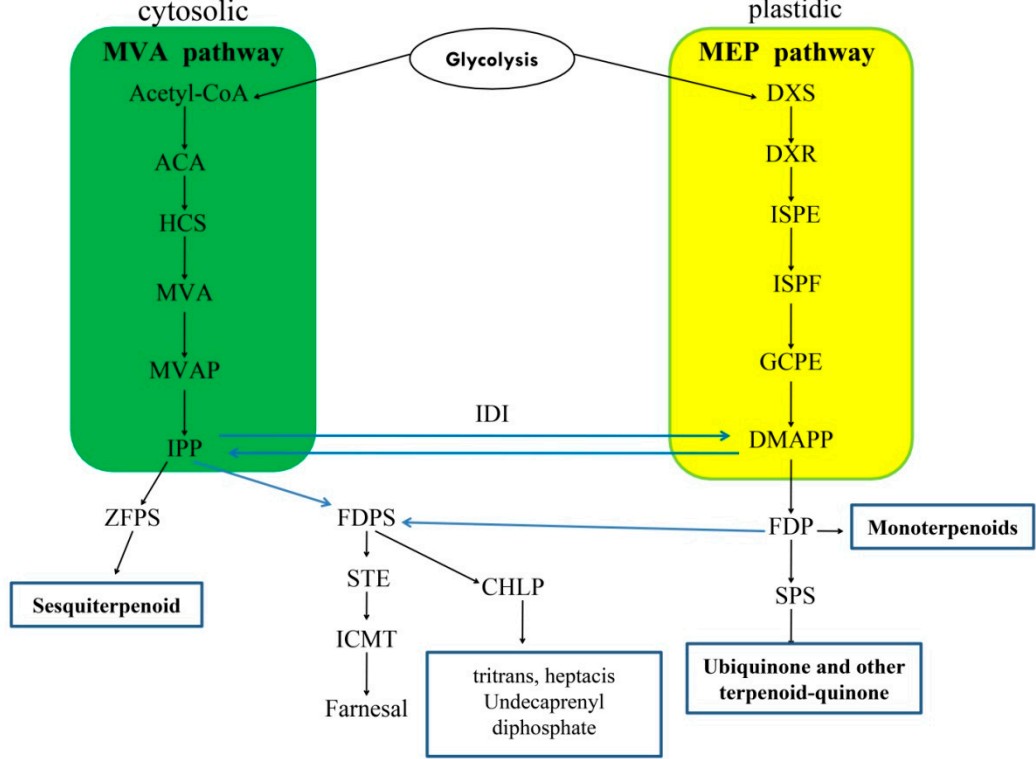

**Figure 1.** Terpenoid biosynthetic pathway in plants. The blue line indicates the part where the two paths intersect. The abbreviationsin the figure: ACA: acetyl-CoA C-acetyltransferase; HCS: hydroxymethylglutaryl-CoA synthase; MVA: mevalonate; MVAP: phosphomevalonate kinase; IPP: isopentenyl pyrophosphate; ZFPS: (2Z,6Z)-farnesyl diphosphate synthase; FDPS: farnesyl diphosphate synthase; STE: STE24 endopeptidase; ICMT: protein-S-isoprenylcysteine O-methyltransferase; CHLP: geranylgeranyl diphosphate/geranylgeranyl-bacteriochlorophyllide a reductase; SPS: all-trans-nonaprenyl-diphosphate synthase; FDP: farnesyl diphosphate synthase; IDI: isopentenyl-diphosphate Delta-isomerase; DMAPP: dimethylally pyrophosphate; GCPE: (E)-4-hydroxy-3-methylbut-2-enyl-diphosphate synthase; ISPE: 4-diphosphocytidyl-2-C-methyl-D-erythritol kinase; ISPF: 2-C-methyl-D-erythritol 2,4-cyclodiphosphate synthase; DXS: 1-deoxy-D-xylulose-5-phosphate synthase; DXR: 1-deoxy-D-xylulose-5-phosphate reductoisomerase.

## 2. Materials and Methods

### 2.1. Materials

Fruit samples of *Z. bungeanum* were obtained from Research Centre for Engineering and Technology of *Zanthoxylum*, State Forestry Administration, Northwest A&F University in Fengxian, Shaanxi Province, China (located in southwest Shaanxi province). *Z. bungeanum* was grown under the same conditions. *Z. bungeanum* was obtained from 5-year-old trees, which have been fruitful for 3 consecutive years. Furthermore, the cultivation management for each tree was basically the same. Fruit of Hanchengdahongpao and Fuguhuajiao were collected at five different developmental stages: stage 1 (5 d after flowering, young fruit, the average diameter is 2.57 mm), stage 2 (30 d after flowering, enlarging fruit, the average diameter is 3.18 mm), stage 3 (55 d after flowering, green mature fruit, the average diameter is 4.53 mm), stage 4 (80 d after flowering, half-red fruit, the average diameter is

4.86 mm), stage 5 (95 d after flowering, full-red fruit, the average diameter is 5.79 mm). In each stage, 200–300 fruits were collected from each tree; fruits of three random trees were collected as biological repeats. During sampling, fruit samples were quickly peeled and the peels were flash frozen in liquid nitrogen and stored at −80 °C until use.

### 2.2. Volatile Organic Compounds Analysis by HS-SPME–GC-MS

Headspace solid-phase micro extraction (Agilent, Santa Clara, CA, USA) was used to collect volatile organic compounds. We quickly ground each sample with liquid nitrogen. Then, 10 μL of normal hexanes was added as an internal standard to 0.1 g of ground tissue to each sample. This was subsequently covered with an aluminum cover. This was placed on a 65 μm PDMS/DVB fiber before this mixture was maintained at 60 °C for 30 min to extract volatile components. Before GC-MS, we placed the sample bottle in the sample inlet, which was kept at 230 °C for 1 min. Thermo Fisher Scientific ISQ&TRACE1310 GC-MS was used for the analysis and detection of VOCs. VOCs were separated using the TG-5MS column (30 m × 0.25 mm, 0.25 μm; Thermo Scientific, Waltham, MA, USA). The procedure used in the experiment is described as follows [4]: kept at an initial temperature of 50 °C for 2 min; ramped up to 130 °C at 3 °C min$^{-1}$ for 2 min; raised the temperature at a rate of 4 °C min$^{-1}$ to 200 °C for 2 min; and increased the temperature at 20 °C min$^{-1}$ to 230 °C for 5 min. Helium was used as a carrier gas at a flow rate of 80.0 mL min$^{-1}$ and the syringe temperature was maintained at 230 °C. The MS conditions included an ionization energy of 70 eV and scan massrange of 50–550 *m/z*. We used the mass spectral library of the NIST to identify VOCs. All experiments were performed in triplicate.

### 2.3. RNA of Fruit Extraction and cDNA Synthesis

The Ominiplant RNA Kit (CWBIO, Beijing, China) was used to extract RNA from *Z. bungeanum* following the manufacturer's instructions. The purification and concentration of RNA were determined using the ultra-micro nucleic acid analyzer (Nano200, AoSheng, Hangzhou, China). We used the 5 × All-In-One RT MasterMix (ABM, Vancouver, Canada) for reverse transcription to generate cDNA following the manufacturer's protocol. The reaction volume of cDNA synthesis was 20 μL, which included: total RNA (variable), 4 μL of 5 × All-In-One RT MasterMix and Nuclease-free H$_2$O (added until 20 μL). The components were thoroughly mixed, collected by brief centrifugation and incubated at 25 °C for 1 min. We incubated this mixture at 42 °C for 15 min, stopped the reaction using 85 °C for 5 min and finally froze it at −20 °C for later use.

### 2.4. Primer Design and Real-Time Quantitative PCR

Primer Premier 5.0 (Palo Alto, CA, USA) was used to design primers. All primer sequences are listed in (Table 1). All candidate genes were selected from the terpenoid biosynthetic pathways in plants (Figure 1). We employed iQ5 real-time quantitative PCR (Bio-Rad, Hercules, CA, USA) to explore the expression profiles of candidate genes. In this experiment, *UBQ* and *TIF* were used as housekeeping reference genes as in a previous study these genes were stably expressed during different developmental stages of *Z. bungeanum* fruit [2]. The reaction included: 10 μL of 2 × SYBR Green Mix (Bioer Technology, Hangzhou, China); 5 μL of cDNA template; 0.6 μL of Forward Primer; 0.6 μL of Reverse Primer; and 3.8 μL of DEPE water. The amplification procedure started with 95 °C for one cycle of 3 min. This was followed by 40 cycles of: 95 °C for 15 s; 60 °C for 15 s; and 72 °C for 30 s. The reaction was terminated at 95 °C for 10 s and 65 °C for 5 s. There was three biological repetitions for each template. The relative mRNA abundance was calculated using the formula: $2^{-\Delta\Delta Ct}$.

Table 1. RT-qPCR primers used in the experiment.

| Gene | Description | Forward Primer (5′-3′) | Reverse Primer (5′-3′) |
|------|-------------|------------------------|------------------------|
| UBQ | ubiquitin extension protein | TCGAAGATGGCCGTACATTG | TCCTCTAAGCCTCAGCACCA |
| TIF | Translation initiation factor | TTCCTCCCATTACGTTGCT | GCTGGTTACGGACTCTTTG |
| DXS | 1-deoxy-D-xylulose-5-phosphate synthase | GAGATTGGGAAGGGAAGAAT | CATCAGCCACAGTCACAGAA |
| HCS | hydroxymethylglutaryl-CoA synthase | AATGTTGCTGGGAAGTTGAA | GGCTACTGTCTTTGCTCGTTA |
| HMGR | hydroxymethylglutaryl-CoA reductase | GTTGACTCCTTGTACCGAAGAT | GCAACTTCTGGCGATACTGA |
| ZFPS | (2Z,6Z)-farnesyl diphosphate synthase | TGGACTTACACTTGCCC | CGAGAGTTTCTTCTTTGG |
| SPS | all-trans-nonaprenyl-diphosphate synthase | CAAAGCATCGTCGTTTAGCC | TTTCCCTCTTCGCATGTCAC |
| ICMT | protein-S-isoprenylcysteine O-methyltransferase | TAATGCTGTGTAACCCCA | CGTAACCCAAAAAACTCTCT |
| IDI | isopentenyl-diphosphate Delta-isomerase | AACACTAACCCAAACCCTG | GCTTCAAACCTTCTTCTCCA |
| FDP | farnesyl diphosphate synthase | TGGACTTACACTTGCCC | CGAGAGTTTCTTCTTTGG |
| ISPE | 4-diphosphocytidyl-2-C-methyl-D-erythritol kinase | GTTCATTTTAGGAGGACCC | CCCATCTTCTCTCTTGTTT |
| ISPF | 2-C-methyl-D-erythritol 2,4-cyclodiphosphate synthase | TCCATCCAACCGTCCAAA | CCCCAATGCTCCCAAAAT |
| FDPS | farnesyl diphosphate synthase | GGAAGCCGACGAACCACA | GCTCTAACTACCCGCGCA |
| GCPE | (E)-4-hydroxy-3-methylbut-2-enyl-diphosphate synthase | CGCAGAGATACGAGAGAAAA | CTCCACCAACATACCCAAAG |
| DXR | 1-deoxy-D-xylulose-5-phosphate reductoisomerase | AGATTATGGCTGGGGAAC | AGAGGAAGGACAAAAGGA |
| STE | STE24 endopeptidase | CGTGCTGGTCTTGTGAAA | GGGCGGATGAGAATAGTG |
| CHLP | geranylgeranyl diphosphate/geranylgeranyl-bacteriochlorophyllide a reductase | GTTGATGGGTATTGATAGGGC | CGTGAAGATGTTTTAGGGTTTT |
| ACA | acetyl-CoA C-acetyltransferase | GCAGCATTGGTTCTAGTGAGTG | GCATCGGCATATCCTGTGAT |

*2.5. Bioinformatics Analysis of Key Genes*

The bioinformatics analysis of key terpenoid synthetic genes used various bioinformatics software. Prot Param (https://web.expasy.org/protparam/) was used to predict protein physicochemical properties. Signal 4.1 analysis was used for signal peptide analysis. Prot Scale was used for hydrophobicity. Phylogenetic analysis was performed using MEGA6.0 software (Center for Evolutionary Medicine and Informatics, Tempe, AZ, USA).

## 3. Results

*3.1. Analysis of Volatile Organic Compounds in Two Cultivars of Z. bungeanum in Different Periods*

The main volatile organic compounds in *Z. bungeanum* were terpenoids, alcohols and esters. A total of 39 substances were detected across the five, which included 26 types of terpenoids, 9 types of alcohols and 4 types of esters (Table 2). α-Pinene had the highest content of the volatile components in the two types of *Z. bungeanum*, which reached the highest values of 24.74% and 20.78%, respectively. Hanchengdahongpao has more volatile compounds than Fuguhuajiao as 35 types of volatile components were detected in Hanchengdahongpao, including 24 types of terpenoids, 7 types of alcohols and 4 types of esters. A total of 21 types of volatile components were detected in Fuguhuajiao, including 16 types of terpenoids, 4 types of alcohols and 1 ester. These volatile components are unique to Hanchengdahongpao: Bicyclo[2.2.1]heptane, D-Limonene, Cyclohexene, Eucalyptol, α-Myrcene, 3-Cyclohexen-1-one, 2-Cyclohexen-1-one, 6-Octenal, GermacreneD, Geraniol, Cyclohexanol, 5-Isopropyl-2-methyl, bicyclo[3.1.0]hexan-2-ol, 3-Cyclohexen-1-ol, 6-Octen-1-ol, Linalyl acetate, Myrtenyl acetate and α-Terpinyl acetate. These volatiles are unique to Fuguhuajiao: (S,1Z,6Z)-8-Isopropyl-1-methyl-5-methyle-necyclodeca-1,6-diene, (1S,4aR,8aS)-1,4a-Isopropyl-7-methyl-4-methylene-1,2,3,4,5,6,8a-octahydronaphthalene, (3E,5E)-2,6-Dimethylocta-3,5,7-trien-2-ol and (2E,4S,7E)-4-Isopropyl-1,7-dimethylcyclodeca-2,7-dienol. In the case of terpenoids, the monoterpenoids were the dominant type of terpenoids (Table 2). A total of 18 types of monoterpene and 6 types of sesquiterpene were released by Hanchengdahongpao, with (1S,2E,6E,10R)-3,7,11,11-Tetramethylbicyclo[8.1.0]undeca-2,6-diene having the highest content among all sesquiterpenes in the two cultivars of *Z. bungeanum.* The heatmap for the content of terpenoids in five different periods is shown in Figure 2a. In general, as the fruit grows, the content of volatile terpenes in *Z. bungeanum* first increases before decreasing.

**Table 2.** Composition and contents of volatile organic compounds in different periods of two cultivars of *Z. bungeanum*.

| Compounds | H1 | H2 | H3 | H4 | H5 | F1 | F2 | F3 | F4 | F5 |
|---|---|---|---|---|---|---|---|---|---|---|
| Monoterpene | | | | | | | | | | |
| α-Pinene | 11.20 ± 0.08 | 12.35 ± 0.12 | 14.31 ± 0.03 | 10.68 ± 0.07 | 9.84 ± 0.03 | 0.96 ± 0.02 | 11.89 ± 0.02 | 13.92 ± 0.02 | 10.75 ± 0.31 | 8.70 ± 0.25 |
| Bicyclo[3.1.0]hexane | UD | UD | 3.36 ± 0.03 | 3.96 ± 0.02 | 1.23 ± 0.02 | 3.27 ± 0.04 | 5.64 ± 0.03 | 7.36 ± 0.01 | 6.75 ± 0.03 | 5.43 ± 0.03 |
| Bicyclo[2.2.1]heptane | UD | UD | 3.94 ± 0.05 | 1.54 ± 0.01 | 1.21 ± 0.01 | UD | UD | UD | UD | UD |
| Bicyclo[3.1.0]hex-2-ene | UD | UD | 1.66 ± 0.02 | 1.75 ± 0.04 | 2.64 ± 0.03 | UD | UD | UD | 0.76 ± 0.04 | 0.93 ± 0.01 |
| Bicyclo[3.1.1]hept-2-ene | UD | UD | 7.11 ± 0.02 | 6.58 ± 0.01 | 2.28 ± 0.03 | 1.06 ± 0.03 | 1.80 ± 0.04 | 8.14 ± 0.02 | 3.48 ± 0.01 | 2.20 ± 0.04 |
| α-Ocimene | 8.20 ± 0.16 | 8.68 ± 0.05 | 8.84 ± 0.01 | 7.21 ± 0.03 | 7.08 ± 0.07 | 6.77 ± 0.04 | 8.85 ± 0.05 | 9.92 ± 0.02 | 8.36 ± 0.42 | 7.18 ± 0.03 |
| Linalool | 1.19 ± 0.01 | 5.38 ± 0.02 | 5.95 ± 0.03 | 2.64 ± 0.03 | 1.78 ± 0.02 | 5.45 ± 0.03 | 6.84 ± 0.03 | 6.92 ± 0.04 | 6.63 ± 0.01 | 6.25 ± 0.07 |
| 2,4,6-Octatriene | UD | UD | UD | 2.38 ± 0.01 | 1.95 ± 0.03 | UD | UD | 1.54 ± 0.03 | 2.28 ± 0.18 | 1.15 ± 0.22 |
| 1,5-Cyclodecadiene | UD | UD | UD | 3.74 ± 0.04 | 1.27 ± 0.03 | UD | UD | 2.78 ± 0.09 | 2.96 ± 0.04 | 0.76 ± 0.01 |
| Terpineol | UD | 2.38 ± 0.02 | 5.45 ± 0.02 | 3.56 ± 0.04 | 1.64 ± 0.02 | UD | UD | 1.04 ± 0.02 | 0.85 ± 0.01 | 0.66 ± 0.01 |
| 1,3-Cyclohexadiene | UD | UD | UD | UD | 0.96 ± 0.01 | UD | UD | UD | UD | 3.11 ± 0.03 |
| D-Limonene | UD | 0.64 ± 0.02 | 3.97 ± 0.01 | 2.48 ± 0.01 | 1.03 ± 0.02 | UD | UD | UD | UD | UD |
| Cyclohexene | UD | UD | UD | 4.17 ± 0.01 | 6.17 ± 0.02 | UD | UD | UD | UD | UD |
| Eucalyptol | 1.25 ± 0.01 | 2.14 ± 0.02 | 2.38 ± 0.02 | 2.48 ± 0.03 | 1.29 ± 0.01 | UD | UD | UD | UD | UD |
| α-Myrcene | UD | UD | 6.29 ± 0.02 | 5.27 ± 0.01 | 1.77 ± 0.02 | UD | UD | UD | UD | UD |
| 3-Cyclohexen-1-one | UD | UD | 3.90 ± 0.04 | 3.22 ± 0.03 | 1.78 ± 0.01 | UD | UD | UD | UD | UD |
| 2-Cyclohexen-1-one | UD | UD | UD | UD | 2.87 ± 0.01 | UD | UD | UD | UD | UD |
| 6-Octenal | UD | UD | UD | UD | 1.51 ± 0.02 | UD | UD | UD | UD | UD |
| Sesquiterpene | | | | | | | | | | |
| Caryophyllene | UD | 1.67 ± 0.08 | 2.66 ± 0.08 | 2.23 ± 0.01 | 0.86 ± 0.04 | 2.55 ± 0.05 | 2.80 ± 0.03 | 1.30 ± 0.03 | 0.94 ± 0.02 | 0.74 ± 0.06 |
| ç-Elemene | UD | UD | UD | UD | 1.04 ± 0.01 | UD | UD | UD | 0.69 ± 0.04 | 0.70 ± 0.07 |
| Humulene | UD | UD | UD | UD | 1.45 ± 0.01 | UD | UD | UD | UD | 0.69 ± 0.02 |
| (1S,2E,6E,10R)-3,7,11,11-Tetramethylbicyclo[8.1.0] undeca-2,6-diene | 1.33 ± 0.04 | 3.13 ± 0.42 | 2.37 ± 0.09 | 2.01 ± 0.03 | 1.35 ± 0.07 | UD | 2.24 ± 0.01 | 7.28 ± 0.02 | 3.28 ± 0.02 | 2.91 ± 0.01 |
| (S,1Z,6Z)-8-Isopropyl-1-methyl-5-methylenecyclodeca-1,6-diene | UD | UD | UD | UD | UD | UD | UD | 0.99 ± 0.02 | 1.03 ± 0.02 | 0.53 ± 0.03 |
| Germacrene D | UD | UD | 3.18 ± 0.01 | 2.14 ± 0.02 | 0.30 ± 0.04 | UD | UD | UD | UD | UD |
| 1-Isopropyl-4,7-dimethyl-1,2,3,5,6,8a-hexahydronaphthalene | UD | UD | UD | UD | 1.03 ± 0.02 | UD | UD | 1.11 ± 0.02 | 1.09 ± 0.01 | 0.63 ± 0.01 |
| (1S,4aR,8aS)-1,4a-Isopropyl-7-methyl-4-methylene-1,2,3,4,5,6,8a-octahydronaphthalene | UD | UD | UD | UD | UD | UD | UD | UD | UD | 0.64 ± 0.04 |
| Alcohols | | | | | | | | | | |
| (3E,5E)-2,6-Dimethylocta-3,5,7-trien-2-ol | UD | UD | UD | UD | UD | UD | UD | UD | 0.77 ± 0.01 | 0.58 ± 0.03 |

**Table 2.** *Cont.*

| Compounds | H1 | H2 | H3 | H4 | H5 | F1 | F2 | F3 | F4 | F5 |
|---|---|---|---|---|---|---|---|---|---|---|
| (2E,4S,7E)-4-Isopropyl-1,7-dimethylcyclodeca-2,7-dienol | UD | UD | UD | UD | UD | UD | UD | UD | UD | 1.19 ± 0.05 |
| 1,6,10-Dodecatrien-3-ol | UD | UD | 2.77 ± 0.05 | 5.86 ± 0.02 | 1.07 ± 0.03 | UD | UD | UD | 0.94 ± 0.02 | 0.8 ± 0.03 |
| (2E,4S,7E)-4-Isopropyl-1,7-dimethylcyclodeca-2,7-dienol | UD | UD | UD | 0.57 ± 0.07 | 1.50 ± 0.08 | UD | UD | UD | UD | 1.46 ± 0.03 |
| Geraniol | UD | UD | UD | UD | 2.90 ± 0.04 | UD | UD | UD | UD | UD |
| Cyclohexanol | UD | UD | UD | 2.61 ± 0.01 | 3.21 ± 0.02 | UD | UD | UD | UD | UD |
| 5-Isopropyl-2-methylbicyclo[3.1.0]hexan-2-ol | UD | UD | UD | UD | 5.15 ± 0.04 | UD | UD | UD | UD | UD |
| 3-Cyclohexen-1-ol | UD | UD | UD | UD | 7.39 ± 0.10 | UD | UD | UD | UD | UD |
| 6-Octen-1-ol | UD | UD | UD | UD | 1.27 ± 0.01 | UD | UD | UD | UD | UD |
| Esters | | | | | | | | | | |
| Geranyl acetate | UD | 2.33 ± 0.12 | 4.93 ± 0.01 | 4.67 ± 0.01 | 1.03 ± 0.04 | UD | UD | UD | UD | 1.39 ± 0.02 |
| Linalyl acetate | UD | 2.83 ± 0.02 | 5.50 ± 0.08 | 4.54 ± 0.02 | 0.60 ± 0.02 | UD | UD | UD | UD | UD |
| Myrtenyl acetate | UD | UD | UD | UD | 1.59 ± 0.02 | UD | UD | UD | UD | UD |
| α-Terpinyl acetate | UD | UD | 2.95 ± 0.01 | 4.30 ± 0.04 | 3.67 ± 0.02 | UD | UD | UD | UD | UD |

The unit is mg/100g FW (fresh weight). H1–H5 represents five periods of Hanchengdahongpao and F1–F2 represents five periods of Fuguhuajiao. UD means Undetected. Data were represented as means ± standard deviations ($n = 3$); values > 0.5% were analyzed.

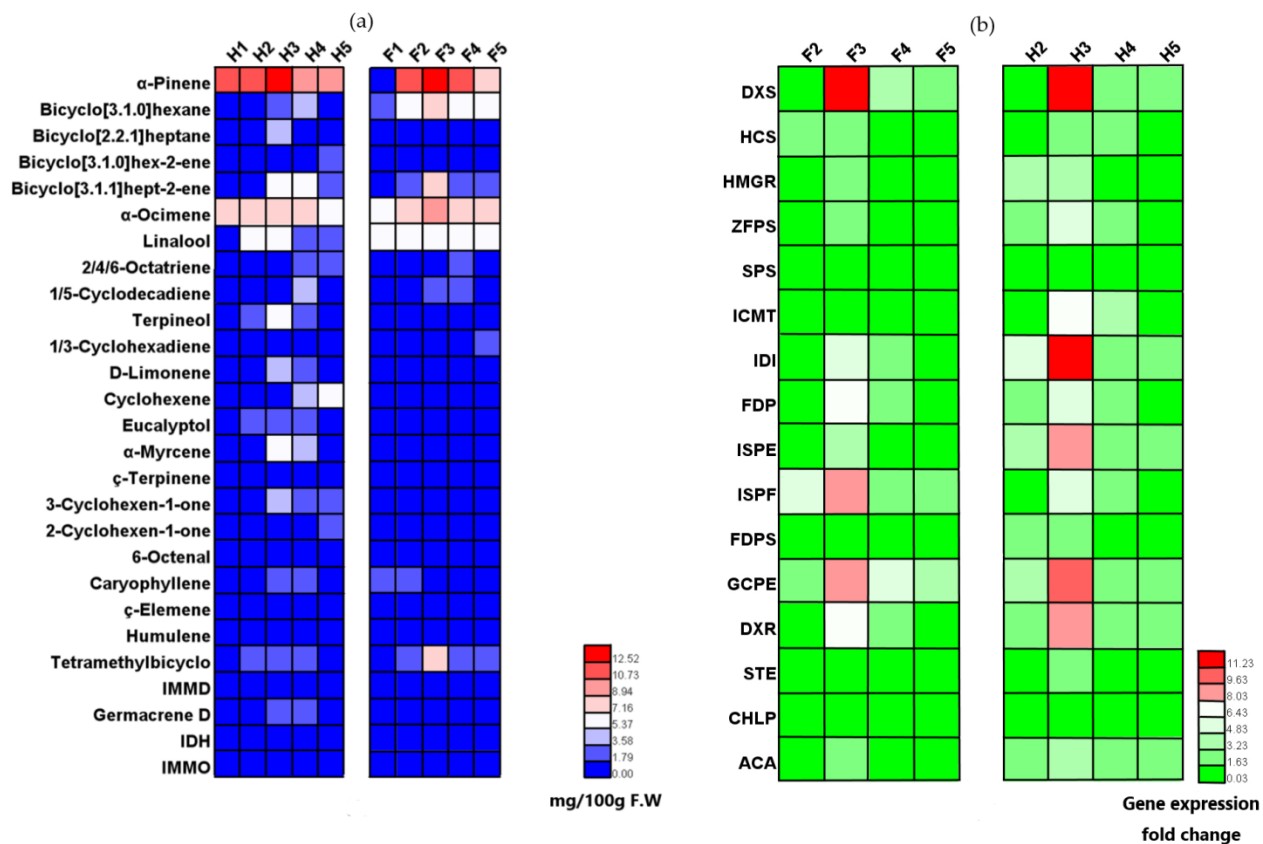

**Figure 2.** Heat map of the content of terpenoids in five different periods of *Z. bungeanum*. H1–H5 represents the five periods of Hanchengdahongpao and F1–F5 represents the five periods of Fuguhuajiao. The abbreviationsin the figure: Tetramethylbicyclo: (1S,2E,6E,10R)-3,7,11,11-Tetramethylbicyclo[8.1.0]undeca-2,6-diene; IMMD: (S,1Z,6Z)-8-Isopropyl-1-methyl-5-methylenecyclodeca-1,6-diene; IDH: 1-Isopropyl-4,7-dimethyl-1,2,3,5,6,8a-hexahydronaphthalene; and IMMO: (1S,4aR,8aS)-1,4a-Isopropyl-7-methyl-4-methylene-1,2,3,4,5,6,8a-octah-ydronaphthalene. The unit is mg/100 g FW (fresh weight). The legend indicates color intensity; (**a**) heat expression of relative expression of terpenoid metabolic pathway candidate genes (*DXS, HCS, HMGR, ZFPS, SPS, ICMT, IDI, FDP, ISPE, ISPF, FDPS, GCPE, DXR, STE, CHLP* and *ACA*) in *Z. bungeanum.* F2–F5 represents the different developmental stages of Fuguhuajiao from development to maturity and H2–H5 represents the different developmental stages of Hanchengdahongpao from development to maturity (**b**).

### 3.2. Expression Profiles of Candidate Genes in Terpenoid Biosynthetic Pathway

In the plants, volatile terpenes originate from the MEP and the MVA pathways. In this study, 16 candidate genes were selected. In order to evaluate and compare their expression levels in fruits at five periods between the two cultivars, the heatmap for gene expression is shown in Figure 2b, which uses F1 and H1 as control groups. The changes in the relative expression of *DXS*, *HCS*, *HMGR*, *ZFPS*, *IDI*, *FDP*, *ISPE*, *ISPF*, *GCPE*, *DXR*, and *ACA* were consistent with the changes in physiological indexes as they all increased during period 1 and 2, reached the peak level of expression in the third period and then began to decline (Table S1, Supplementary Materials). According to the heatmap, among these genes, the most significant upregulation gene is *DXS* (its relative expression was 12 times higher than the control group), which is also an important rate-limiting enzyme in the MEP pathway. In contrast, the key rate-limiting enzyme *HMGR* on the MVA pathway has a lower relative expression (three times higher than the control group). The relative expression levels of *SPS*, *ICMT*, *FDPS*, *STE*, and *CHLP* are low or down regulated.

### 3.2.1. Phylogenetic Analysis of *ZbDXS*

In order to determine the evolutionary status of *DXS*, we built a phylogenetic tree using MEGA6.0 with the neighbor-joining method (Figure 3). The amino acid sequence was obtained from the NCBI database (Table 3). According to Figure 3, it can be concluded that the plant *DXS* was clustered into three categories. Clade 1 and clade 3 can be identified based on the well-characterized DXS1 proteins of Arabidopsis and *M. truncatula* and DXS3 protein of Arabidopsis [31]. Clade 2 can be identified based on DXS2 protein of *Taxus × media.* Numbers of Figure 3 represented the genetic distance; the genetic distance between *ZbDXS* and *CitDXS* was the smallest.

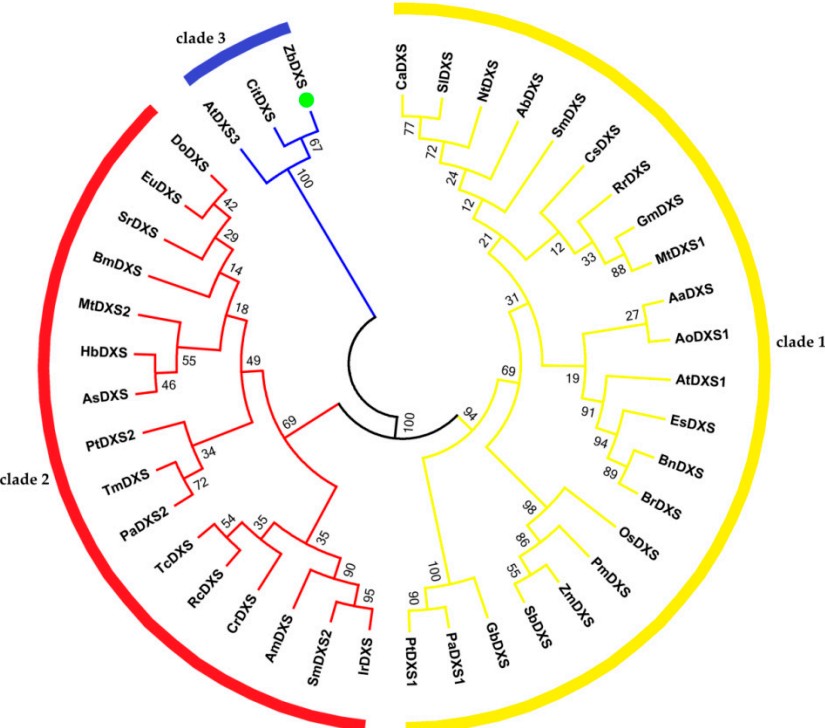

**Figure 3.** Phylogenetic tree of *DXS* from *Z. bungeanum* and other plants. The *DXS* of *Z. bungeanum* is marked with a green dot. Different colors were used to distinguish different clades. Yellow represents clade 1, red represents clade 2 and blue represents clade 3.

### 3.2.2. Bioinformatics Analysis of *ZbDXS* of *Z. bungeanum*

The analysis of the amino acid sequence of the *ZbDXS* gene using Prot Param revealed that it has 256 amino acids; a molecular weight of 28, 191.55 Da; theoretical pI of 5.50; 34 negatively charged residues (Asp + Glu); 29 positively charged residues (Arg + Lys); molecular formula of $C_{1228}H_{1952}N_{344}O_{400}S_8$; 3932 atoms; an instability index(II) of 30.36; aliphatic index of 74.37; and grand average of hydropathicity (GRAVY) of −0.588. We analyzed the hydrophobicity of the *ZbDXS* amino acid sequence with the Prot Scale (Figure 4). According to the data in the figure, we obtained the following conclusions. In the *ZbDXS* polypeptide chain, hydrophilic amino acids are uniformly distributed in a greater amount than hydrophobic amino acids. Therefore, the entire *ZbDXS* peptide chain shows a certain hydrophilicity. The signal peptide is located at the N-terminus of the secreted protein. It usually consists of 15 to 30 amino acids. We analyzed *ZbDXS* amino acids using the signal 4.1 server to sequence the signal peptide and its position (Figure 5). The maximum value of the raw cleavage site score occurs at the 56th amino acid and the value is 0.112. The maximum value of the combined cleavage site score occurs at the 68th amino acid with a score of 0.107. The maximum value of the single peptide score occurs at the 50th amino acid with a score of 0.119. The average signal peptide S-value is between the 1st and 67th amino acids and its value is 0.096. Based on the above data, we concluded that *ZbDXS* has no signal peptide present and it is likely that it is not a secreted protein.

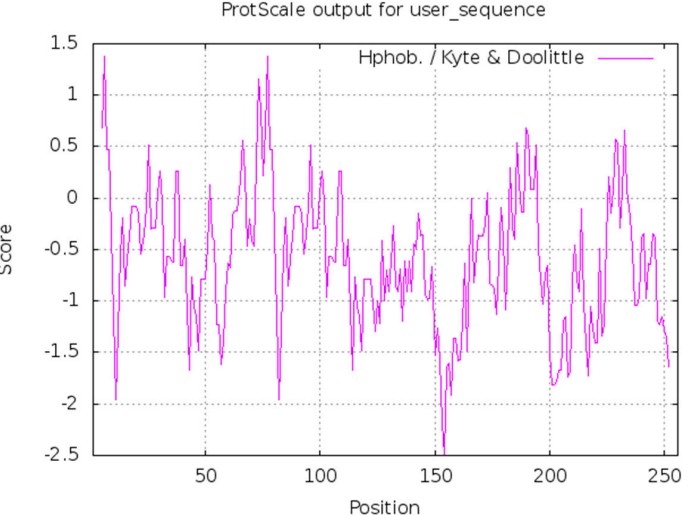

**Figure 4.** *ZbDXS* amino acid sequence hydrophilicity.

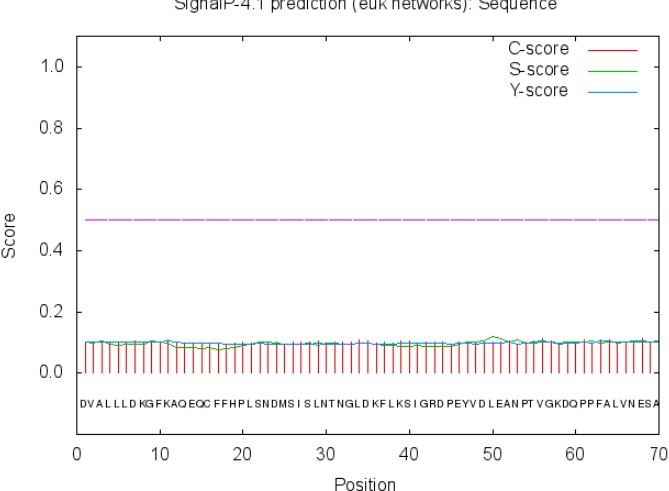

**Figure 5.** *ZbDXS* amino acid sequence signal peptide analysis.

**Table 3.** Other plant DXS proteins used for phylogenetic tree analysis.

| Protein | Protein ID in NCBI | NOTE |
|---|---|---|
| CitDXS | XP_006483309.1 | *Citrus sinensis* |
| AtDXS1 | AEE76517.1 | *Arabidopsis thaliana* |
| RrDXS | AEZ53173.1 | *Rosa rugosa* |
| GmDXS | NP_001236070.1 | *Glycine max* |
| ZmDXS | ACT32136.1 | *Zea mays* |
| AsDXS | AHI62962.1 | *Aquilaria sinensis* |
| IrDXS | AMM72794.1 | *Isodonrubescens* |
| NtDXS | CBA12009.1 | *Nicotiana tabacum* |
| SmDXS | ACF21004.1 | *Salvia miltiorrhiza* |
| AbDXS | AGG54706.1 | *Aconitum balfourii* |
| BnDXS | AHN09416.1 | *Brassica napus* |
| OsDXS | XP_015640505.1 | *Oryza sativa* |
| CrDXS | ABI35993.1 | *Catharanthus roseus* |
| GbDXS | AAS89341.1 | *Ginkgo biloba* |
| AaDXS | AAD56390.2 | *Artemisia annua* |
| EuDXS | AFU93069.1 | *Eucommia ulmoides)* |
| CaDXS | PHT95029.1 | *Capsicum annuum* |
| SlDXS | CAZ66648.1 | *Solanum lycopersicum* |
| SbDXS | XP_002441088.1 | *Sorghum bicolor* |
| EsDXS | XP_006414485.1 | *Eutremasalsugineum* |
| AoDXS1 | AEK69518.1 | *Alpiniaofficinarum* |
| SmDXS2 | ACQ66107.1 | *Salvia miltiorrhiza* |
| AmDXS | AAW28999.1 | *Antirrhinum majus* |
| AtDXS3 | AED91669.1 | *Arabidopsis thaliana* |
| BrDXS | ABE60813.1 | *Brassica rapa* |
| HbDXS | ABF18929.1 | *Heveabrasiliensis* |
| MtDXS1 | CAD22530.1 | *Medicago truncatula* |
| MtDXS2 | CAN89181.1 | *Medicago truncatula* |
| PaDXS1 | ABS50518.1 | *Piceaabies* |
| PaDXS2 | ABS50520.1 | *Piceaabies* |
| PtDXS1 | ACJ67021.1 | *Pinus taeda* |
| PtDXS2 | ACJ67020.1 | *Pinus taeda* |
| RcDXS | XP_002533688.1 | *Ricinus communis* |
| SrDXS | ACI43010.1 | *Stevia rebaudiana* |
| TmDXS | AAS89342.1 | *Taxus×media* |
| DoDXS | AHF22384.1 | *Dendrobium officinale* |
| CsDXS | BAF75640.1 | *Croton stellatopilosus* |
| TcDXS | EOY31729.1 | *Theobroma cacao* |
| BmDXS | ACP30544.2 | *Bacopa monnieri* |
| PmDXS | RLN29936.1 | *Panicum miliaceum* |

## 4. Discussion

Terpenoids are one of the most important secondary metabolites in plants [8], which strongly contribute to the unique flavor of *Z. bungeanum*. In this study, α-Pinene was detected as having the highest content out of all the VOCs, which suggests that it might be an important internal factor determining the flavor of *Z. bungeanum*. Related research shows that α-Pinene has good therapeutic effects on some diseases [32]. There needs to be further research on the use of aroma as pharmacology.

According to previous studies and the results of this experiment [14], we found that *DXS* plays an important role in the synthesis of *Z. bungeanum* aroma. The *DXS* family is known to be very small with three subfamilies [33]. The *DXS1* family mainly plays the role of housekeeping genes while *DXS2* and *DXS3* are mainly involved in plant secondary metabolism [31]. Phylogenetic analysis found that the genetic distance between *ZbDXS* and the *DXS* gene (*CitDXS*) of *Citrus sinensis* was the smallest. *ZbDXS* and *AtDXS3* (*Arabidopsis thaliana*) were divided into one clade, and *AtDXS3* belongs to the *DXS3* type gene. Thus, it is speculated that *ZbDXS* is also likely to be of the *DXS3* type. It is currently

known that *DXS3* family has the fewest member. The expression level of *DXS3* in maize was low in different tissues, it was speculated that this gene was mainly involved in MEP pathways with lower levels, such as phytohormone gibberellin and abscisic acid [34]. At present, there is no report on the *DXS* gene family in *Z. bungeanum*, which will be needed to further understand the mechanism related to aroma.

The results of RT-qPCR showed that compared to the genes in the MVA pathway, the key genes in the MEP pathway have relatively strong changes in their relative expression. This result presumes that the volatile terpenoids in *Z. bungeanum* are mainly obtained from the MEP pathway. The results of GC-MS also support this view. The MEP pathway is widely found in plants, most eukaryotic bacteria and some parasites, which can form monoterpenes, diterpenes and tetraterpenes in plants [35].

The aroma components of fruits vary greatly among different species and there are differences in the aroma between different varieties of the same species due to genotype differences. There are two main types of *Z. bungeanum*, which are namely *Zanthoxylum bungeanum* (*Z. bungeanum*) and *Zanthoxylum armatum* (*Z. armatum*) [2]. The aroma of *Z. armatum* is more intense than *Z. bungeanum*. The two types of *Z. bungeanum* have different aroma types, which are determined by their environmental factors and genes. In general, the aroma of *Z. armatum* belongs to the fragrance type, while the *Z. bungeanum* belongs to the rich aroma. Different species have different aroma changes. *Fragaria × ananassa* Duch. is an important plant for studying the biosynthesis of aroma substances during growth and development as the content of its alcohol gradually decreases during the growth process [36].

During the maturation process of *Z. bungeanum*, the aroma substances began to gradually synthesize. Thus, the related key genes and corresponding regulatory factors should be a main focus of future research.

In addition to genes, the production of volatile terpenoids is influenced by many factors, such as environment, hormones, etc. Skinkis found that grapefruit significantly increased the content of bound terpenes in daylight, but this had no effect on the free terpenes [37]. Toselli's results showed that the linalool content of the *Prunus persica* var after applying organic fertilizer was higher [38]. The aroma of *Vitis vinifera* L. has the highest content under mild water stress and moderate nitrogen supply while severe water stress and nitrogen deficiency will weaken the aroma [39]. The analysis of the aroma of *Malus domestica* found that the esters were considered to be the major contributors to most cultivars of *Malus domestica*. The biosynthesis of ester volatiles during *Malus domestica* ripening requires not only sustained ethylene but also a high ethylene yield [40]. For *Z. bungeanum*, the aroma can be modified by treating the environment and hormones, which will be an important direction of further research.

## 5. Conclusions

In this study, the changes in volatile terpenoids in *Z. bungeanum* were determined as described. As the fruit develops, volatile terpenoids continue to accumulate until they reach their peak. At this point, any further development results in the content of volatile terpenoids beginning to decline but it remains higher than the levels in the early stages of development (the first period). The composition of volatile terpenoids of the two cultivars in the five periods was different. Hanchengdahongpao has more volatile terpenoids than Fuguhuajiao. Furthermore, α-pinene is the most abundant terpene in the two cultivars, reaching 24.74% and 20.78% in Hanchengdahongpao and Fuguhuajiao, respectively. A total of 39 VOCs were detected in five periods, which were mainly terpenoids. A total of 35 types of VOCs were detected in Hanchengdahongpao, including 24 volatile terpenoids. Fuguhuajiao has 16 types of VOCs, including 11 volatile terpenoids. Monoterpenes are the main type of terpenoids found. We detected 24 volatile terpenoids in Hanchengdahongpao, 18 of which are monoterpenoids. Among the 16 volatile terpenoids in Fuguhuajiao, there are 9 types of monoterpenoids. The results of real-time quantitative PCR were consistent with physiological indicators as most of the genes were upregulated and the change trend of gene expression was consistent with the change involatile terpenoids. Among them, the key rate-limiting enzyme *ZbDXS* in the MEP pathway showed the

strongest upregulation in gene transcript level. The results of the phylogenetic tree of *ZbDXS* show that *DXS* gene family can be clustered into three independent branches, with the closest relationship occurring between *ZbDXS* and *DXS* of *Citrus sinensis*, *CitDXS*.

**Supplementary Materials:** The following are available online at http://www.mdpi.com/1999-4907/10/4/328/s1, Table S1: Relative expression levels of terpenoid metabolic pathway candidate genes in *Z. bungeanum* in fruit during 5 developmental stages.

**Author Contributions:** A.W. conceived the project. J.S. and X.F. designed the experiments and performed the experiment. J.S. wrote the paper. All authors (J.S., X.F., Y.H., Y.L., and A.W.) discussed the results and commented on the manuscript.

**Funding:** This study was financially supported by the National Key Research and Development Program Project Funding (2018YFD1000605).

**Acknowledgments:** We thank Yao Ma for his advice and help for this experiment.

**Conflicts of Interest:** The authors declare no conflict of interest.

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
