# Peer review of "Identification of Key Genes in the Synthesis Pathway of Volatile Terpenoids in Fruit of Zanthoxylum bungeanum Maxim"

_forests, doi:10.3390/f10040328_

Reviewer 1 Report

The paper by Shi and colleagues is aimed to increase understanding of the molecular processes regulating of volatile terpenoids synthesis in Zanthoxylum. On the whole, the purpose of the MS is supported by a depth and good experimental design. However, I consider that some aspects minor points correction should be performed.

Some typos are spread along the text: monkeyflower; gibberellinand; manufacturer'sprotocol; bungeanumare; 

Figure 4 some graphic imperfections: The green dot location; TmDXS is not readable as well as some numbers

Author Response

Response to Reviewer 1 Comments

Point 1: Some typos are spread along the text: monkeyflower; gibberellinand; manufacturer'sprotocol; bungeanumare; . 

Response 1: These words are all missing spaces. monkeyflower modified to monkey flower; gibberellinand modified to gibberellin and; manufacturer'sprotocol modified to manufacturer's protocol; bungeanumare modified to bungeanum are.

Point 2: Figure 4 some graphic imperfections: The green dot location; TmDXS is not readable as well as some numbers

Response 2: There are some errors in this figure, I have refined this figure in the manuscript.

Reviewer 2 Report

This is an interesting study on volatile compounds. This approach would be useful in other fragrant crops, such as roses. The authors performed some interesting and useful analyses, but there are some major issues with the manuscript that need to be addressed. Many of the comments and edits are added in the pdf.

What MUST be addressed before publication is a detailed description of the 5 periods. This needs to be extremely clear so others can replicate this work and understand the significance of the authors' findings.

The RT-qPCR may not have the proper controls. Have the authors verified that ubiquitin is appropriate in this crop for each developmental stage? As fruits mature, is it common for transcript degradation to occur in the later stages of senescence. It is unclear what stages they collected for their experiment so I do not know if this is a problem or not. It is also becoming more common that two housekeeping genes are used to normalize qPCR.

Many of the figures need to be further refined. For example, the phylogenetic tree has many issues and it is unclear why the phylogeny is not reflective of the 3 groups of DXS.

The heatmap figures should be presented in a way that it is easy for the reader to compare the volatile profile with the genetic profile. The in silico amino acid analyses are not very informative for the manuscript nor to the reader. I would replace all of those with a nice developmental photo of the fruit stages at the time of harvest.

Please see the other comments and edits in the pdf.

Author Response

Response to Reviewer 2 Comments

Point 1: What MUST be addressed before publication is a detailed description of the 5 periods. This needs to be extremely clear so others can replicate this work and understand the significance of the authors' findings.

Response 1: Fruit of Hanchengdahongpao and Fuguhuajiao were collected at five different developmental stages: stage 1 (5 d after flowering, young fruit, the average diameter is 2.57 mm), stage 2 (30 d after flowering, enlarging fruit, the average diameter is 3.18 mm), stage 3 (55 d after flowering, green mature fruit, the average diameter is 4.53 mm), stage 4 (80 d after flowering, half-red fuit, the average diameter is 4.86 mm), stage 5 (95 d after flowering, full-red fruit, the average diameter is 5.79 mm). In each stages, 200-300 fruits were collected from each tree, fruits of random three trees were collected as biological repeats.

Point 2: The RT-qPCR may not have the proper controls. Have the authors verified that ubiquitin is appropriate in this crop for each developmental stage? As fruits mature, is it common for transcript degradation to occur in the later stages of senescence. It is unclear what stages they collected for their experiment so I do not know if this is a problem or not. It is also becoming more common that two housekeeping genes are used to normalize qPCR.

Response 2: We reworked the RT-qPCR experiment. This time we used two housekeeping genes (UBQ, TIF) as controls to make the results more reliable. The data is uploaded in Additional file (excel form).

Point 3: Many of the figures need to be further refined. For example, the phylogenetic tree has many issues and it is unclear why the phylogeny is not reflective of the 3 groups of DXS.

Response 3: I have refined some of figures in manuscript: Combining Figure 2 and Figure 3 into one figure, refining the phylogenetic tree, and Figure 7 was removed; DXS1, DXS2 and DXS3 have no direct relationship between their phylogenetic tree classification and the numbering behind their genes [1]. Plant gene family classifications are not necessarily classified by numerical number, and may be classified according to the number of chromosomes or alleles.

Point 4: The heatmap figures should be presented in a way that it is easy for the reader to compare the volatile profile with the genetic profile. The in silico amino acid analyses are not very informative for the manuscript nor to the reader. I would replace all of those with a nice developmental photo of the fruit stages at the time of harvest.

Response 4: I put two heatmap figures together so that readers can compare them better; Analysis of the secondary structure of the amino acid sequence has little significance to the manuscript, so we removed Figure 7; We have already described the stage of fruit development in detail to facilitate replication.

[1] Zhang, F.; Liu, W.; Xia, J.; Zeng, J.; Xiang, L.; Zhu, S.; Zheng, Q.; Xie, H.; Yang, C.; Chen, M.; et al. Molecular Characterization of the 1-Deoxy-d-Xylulose 5-Phosphate Synthase Gene Family in Artemisia annua. Front. Plant Sci. 2018, 9, 952.

Round  2

Reviewer 2 Report

The authors have made significant improvements to the manuscript and addressed the major issues the needed to be addressed. For example, the added description of the stage of fruit collection was great!

Generally, it would be much better if they authors would quantify with numbers, percentages, fold differences, etc. as much as possible throughout the manuscript.

Line 60 a comma is needed before “but”

Line 118 change “random three” to “three random”

Line 120 change “liquid nitrogen stored” to “liquid nitrogen and stored”

Line 143 “Variable” should be lowercase

Figure 2 legend has “Figure 2” twice; this probably needs to be adjusted as per the journal’s style.

On page 10 of 18 section 3.2 the authors write, “…as they all increased first…” change this to say “as they all increased during period X” and fill in the appropriate period at which they increased.

Also in this section they authors write, “…the most intense change in the relative expression of its self-gene is DXS…” The term “self-gene” is used in the next sentence as well. I don’t know what a “self-gene” is. I think the authors need to use a more universally known term. While they are at it, instead of just saying “lower relative expression” it would be much better to quantify this number and other numbers of expression in this section. How much did the expression change? It is always best to quantify with numbers in a paper when possible rather than writing “low,” “high,” and “peak.”

Page 15 of 18 Line 9 Italicize “DXS3” to be, “The expression level of DXS3 in maize was low in different tissues…”

On the same page the authors wrote, “…and corresponding regulatory factors should be the hotspots and focuses of future research.” Revise to “…and corresponding regulatory factors should be a main focus of future research.”

In the next paragraph it states, “…terpenes in daylight but this had no effect…” This needs to have a comma before “but.” When two complete sentences are joined with a conjunction (but, and, or, nor, etc.) a comma is needed before the conjunction.

Author Response

Response to Reviewer 2 Comments

Point 1: Line 60 a comma is needed before “but”.

Response 1: I have corrected it in the manuscript

Point 2: Line 118 change “random three” to “three random”

Response 2: I have corrected it in the manuscript

Point 3: Line 120 change “liquid nitrogen stored” to “liquid nitrogen and stored”

Response 3: I have corrected it in the manuscript

Point 4: Line 143 “Variable” should be lowercase

Response 4: I have corrected it in the manuscript

Point 5: Figure 2 legend has “Figure 2” twice; this probably needs to be adjusted as per the journal’s style.

Response 5: I have corrected it in the manuscript

Point 6: On page 10 of 18 section 3.2 the authors write, “…as they all increased first…” change this to say “as they all increased during period X” and fill in the appropriate period at which they increased.

Response 6: I have corrected it in the manuscript

Point 7: Also in this section they authors write, “…the most intense change in the relative expression of its self-gene is DXS…” The term “self-gene” is used in the next sentence as well. I don’t know what a “self-gene” is. I think the authors need to use a more universally known term. While they are at it, instead of just saying “lower relative expression” it would be much better to quantify this number and other numbers of expression in this section. How much did the expression change? It is always best to quantify with numbers in a paper when possible rather than writing “low,” “high,” and “peak.”

Response 7: In this study, our RT-qPCR was a relatively quantitative experiment, we can got a trend (up or down), cannot be quantified. We can got multiple relationships relative to the control group. It was a relative expression, not an absolute quantification. This section has an incorrectly stated part, I have modified it to “According to the heatmap, among these genes, the most significant upregulation gene is DXS (It’s relative expression was 12 times higher than the control group), which is also an important rate-limiting enzyme in the MEP pathway. In contrast, the key rate-limiting enzyme HMGR on the MVA pathway has a lower relative expression (3 times higher than the control group).

Point 8: Page 15 of 18 Line 9 Italicize “DXS3” to be, “The expression level of DXS3 in maize was low in different tissues…

Response 8: I have corrected it in the manuscript

Point 9: On the same page the authors wrote, “…and corresponding regulatory factors should be the hotspots and focuses of future research.” Revise to “…and corresponding regulatory factors should be a main focus of future research.”

Response 9: I have corrected it in the manuscript

Point 10: In the next paragraph it states, “…terpenes in daylight but this had no effect…” This needs to have a comma before “but.” When two complete sentences are joined with a conjunction (but, and, or, nor, etc.) a comma is needed before the conjunction.

Response 10: I have corrected it in the manuscript
